# MIDAS: MOSAIC INPUT-SPECIFIC DIFFERENTIABLE ARCHITECTURE SEARCH

## ABSTRACT

Differentiable Neural Architecture Search (NAS) provides efficient, gradient-based methods for automatically designing neural networks, yet its adoption remains limited in practice. We present MIDAS, a novel approach that modernizes DARTS by replacing static architecture parameters with dynamic, input-specific parameters computed via self-attention, enriching differentiable NAS with the representational power of self-attention. To improve robustness, MIDAS (i) localizes architecture selection by computing it separately for each spatial patch of the activation map, and (ii) introduces a parameter-free, topology-aware search space that models node connectivity and simplifies selecting the two incoming edges per node. We evaluate MIDAS on the DARTS, NAS-Bench-201, and RDARTS search spaces. In DARTS, it reaches 97.42% top-1 on CIFAR-10 and 83.38% on CIFAR-100. In NAS-Bench-201, it consistently finds globally optimal architectures. In RDARTS, it sets the state of the art on two of four search spaces on CIFAR-10. We further analyze why MIDAS works, showing that patchwise attention improves discrimination among candidate operations, and the resulting input-specific parameter distributions are class-aware and predominantly unimodal, providing reliable guidance for decoding.

## 1 INTRODUCTION

Neural architecture search (NAS) automates the design of neural networks by exploring a vast space of candidate architectures. In differentiable NAS, the discrete search space is relaxed into a continuous one and optimized via gradient descent (Liu et al., 2019). Despite promising results (Wu et al., 2019; Zhang et al., 2021b), broader adoption remains limited by instability (Zela et al.; Yang et al., 2020; Chu et al., 2021). Meanwhile, self-attention–based transformer models have proved effective across various domains (Vaswani et al., 2017; Brown et al., 2020; Dosovitskiy et al., 2021; Radford et al., 2021). However, when memory and compute are constrained, NAS remains a compelling approach for discovering efficient, hardware-aware architectures (Wu et al., 2019). Despite its speed and efficiency, differentiable NAS still requires improvements to be broadly adopted.

To improve its effectiveness, we introduce **Mosaic Input-specific Differentiable Architecture Search (MIDAS)**, which incorporates self-attention into the DARTS (Liu et al., 2019) framework (see Figure 1). The key idea is to reinterpret self-attention as a NAS mechanism, where we dynamically adjust connections and operations for each sample and at each node rather than relying on a single shared architecture. While prior work has explored self-attention *within* the search space as an operation (Nakai et al., 2020; Guan et al., 2021), to the best of our knowledge we are the first to leverage self-attention to *optimize over* the search space itself. The most closely related work, AGNAS (Sun et al., 2022), computes input-specific gating parameters for channel recalibration. In contrast, MIDAS leverages full representational power of self-attention.

AGNAS computes input-specific parameters by first applying global average pooling to the activation maps. However, we observe that this can blur differences among candidate operations due to overly coarse features, particularly in early layers where features are local and spatial dimensions are large. We address this by adopting a *patchwise* design, where all activation maps are divided into $P^2$ spatial patches, and self-attention is applied *independently* within each patch, producing $P^2$ sets of architecture parameters. We call this a *mosaic* approach, since the image-level architecture is composed of patch-level architectures.

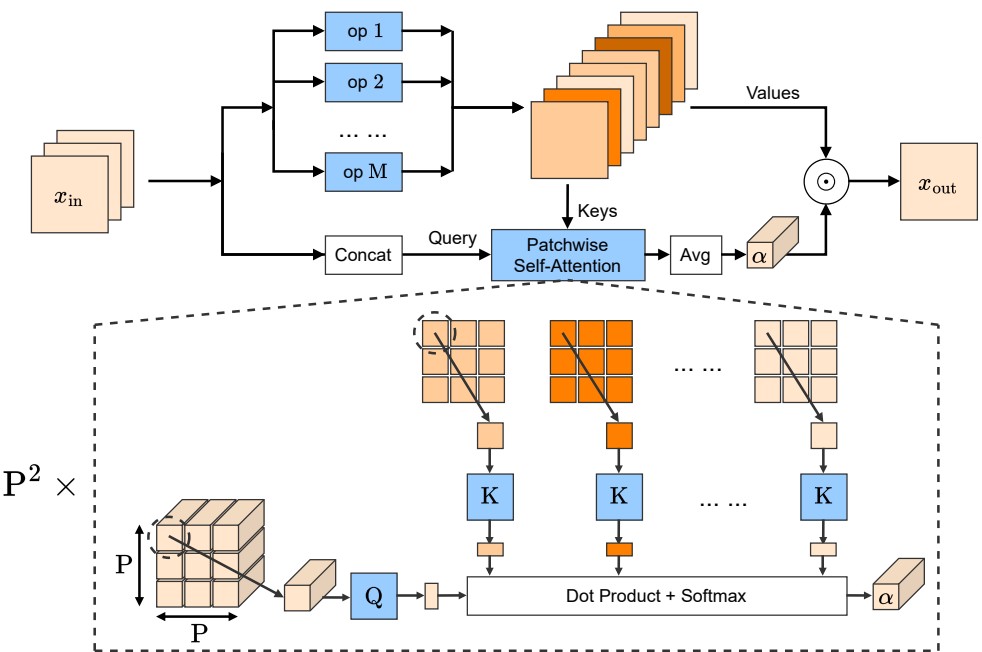

Figure 1: Computing input-specific architecture using self-attention. We use the node's input and the candidate activation maps as the query and keys, respectively. Each activation map is partitioned into $P^2$ patches, and self-attention is applied *separately* within each patch. For clarity, the topology-aware search space is not illustrated.

Another issue in DARTS concerns capturing cell topology using only operation-wise architecture parameters (Wang et al., 2021; Gu et al., 2021), even though the final architecture must select two incoming edges per node. Similar to DOTS (Gu et al., 2021), we consider all possible pairs of candidate edges. However, unlike DOTS, which introduces a separate topological parameter $\beta_{i_1,i_2}$ alongside standard $\alpha$ parameters, we seamlessly integrate topology search into our dynamic, self-attention–based mechanism without adding parameters. This formulation simplifies selection of two incoming edges in the DARTS space and alleviates decoding issues. The final architecture is obtained by averaging input-specific parameters over a subset of the training data and then selecting both operations and connections based on this average.

MIDAS thus combines self-attention–based, input-specific NAS with localized architecture and a parameter-free topology search that accounts for node connectivity. As shown in Section 4, it retains the efficiency of DARTS while leveraging the representational strength of self-attention. In the NAS-Bench-201 search space (Dong & Yang, 2020), MIDAS consistently finds the optimal or near-optimal architecture; in the DARTS search space (Liu et al., 2019), it reaches $97.42\%$ accuracy on CIFAR-10; and on RDARTS S1-S4 (Zela et al.), it is robust across all search spaces and sets state of the art on CIFAR-10 for S2 and S4. We also include additional studies that provide deeper insights into the method.

We summarize our main contributions as follows:

- **MIDAS:** using self-attention to compute input-specific architecture parameters, enabling dynamic per-input architecture selection.

- **Mosaic (patchwise) architecture:** local architecture decisions that better discriminate among candidate operations.

- **Parameter-free topology search:** models node connectivity within the same attention mechanism, avoiding extra parameters.

- **Strong results and analysis:** state-of-the-art or competitive performance across multiple search spaces and datasets, with studies that shed light on MIDAS.

## 2 RELATED WORK

### 2.1 PRELIMINARIES

**DARTS Search Space.** In DARTS (Liu et al., 2019), the *supernet* is a sequence of $N$ convolutional cells, each either a normal or a reduction cell (Zoph et al., 2018). A cell is a densely-connected directed acyclic graph (DAG) with $N$ nodes.

Let $\mathcal{O} = \{o^{(1)}, \dots, o^{(M)}\}$ be the set of $M$ candidate operations. Consider the $k$-th node. For each possible input $x^{(i)} \in \mathbb{R}^{B \times C \times H \times W}$ and operation $o^{(j)} \in \mathcal{O}$, DARTS assigns a learnable parameter $\alpha^{(i,j)}$ (shared across nodes) that indicates the relative importance of applying $o^{(j)}$ to $x^{(i)}$:

$$F^{(i,j)} = o^{(j)}(x^{(i)}), \quad F^{(i,j)} \in \mathbb{R}^{B \times C \times H \times W}. \tag{1}$$

The output of the $k$-th node is a weighted sum over all such candidate activation maps:

$$x_{\text{out}} = \sum_{i=1}^{k+1} \sum_{j=1}^{M} p^{(i,j)} F^{(i,j)}, \tag{2}$$

where

$$p^{(i,j)} = \frac{\exp(\alpha^{(i,j)})}{\sum_{j'=1}^{M} \exp(\alpha^{(i,j')})}. \tag{3}$$

The cell output is obtained by concatenating the outputs of all intermediate nodes.

**Optimization.** Because the supernet is differentiable, convolution weights $\omega$ and architecture parameters $\alpha$ are optimized jointly via a bilevel objective:

$$\min_{\alpha} \mathcal{L}_{\text{A}}(\omega^*, \alpha) \quad \text{s.t.} \quad \omega^* = \arg\min_{\omega} \mathcal{L}_{\text{B}}(\omega, \alpha), \tag{4}$$

where $\mathcal{L}_{\text{A}}$ and $\mathcal{L}_{\text{B}}$ are cross-entropy losses on disjoint splits of the training data A and B. Solving this exactly is expensive, so DARTS relies on an alternating approximation:

- Update architecture parameters $\alpha$ using $\nabla_{\alpha} \mathcal{L}_{\text{A}}(\omega, \alpha)$,
- Update convolution weights $\omega$ using $\nabla_{\omega} \mathcal{L}_{\text{B}}(\omega, \alpha)$.

After convergence, the architecture is discretized by keeping the two strongest (nonzero) operations entering each node.

### 2.2 RANK DISORDER IN DARTS

Although DARTS (Liu et al., 2019) was pivotal in developing differentiable NAS, later work (Zela et al.; Chu et al., 2021; Yu et al., 2020; Gu et al., 2021) revealed several limitations. One of the major shortcomings is *rank disorder*, where operation weights fail to reflect operation's true importance (Yu et al., 2020). DARTS's decoding procedure can exacerbate rank disorder, as simply picking the top-two operations per node based on operation-wise architecture parameters may be misleading. P-DARTS (Chen et al., 2021) partially mitigates it by progressively pruning weaker operations, and DARTS-PT (Wang et al., 2021) keeps only operations whose removal causes the largest decrease in accuracy. DOTS (Gu et al., 2021) observes that edge-level importance is poorly captured by operation-wise parameters and proposes a topology search that assigns distinct parameters to every pair of candidate edges.

### 2.3 INPUT-SPECIFIC NAS

Attention mechanisms emerged in 2014 for RNN-based image classification (Mnih et al., 2014) and have since been applied for various purposes, including channel recalibration and input-specific aggregation (Xu et al., 2015; Hu et al., 2017; Woo et al., 2018). The introduction of the self-attention Transformer (Vaswani et al., 2017) revolutionized sequence modeling, inspiring extensive research on attention-based architectures.

NAS has leveraged attention in several ways. For instance, ADARTS (Xue & Qin, 2023) employs attention to identify the most relevant channels during the search, improving efficiency of the supernet. Other methods directly search for attention modules (Guan et al., 2021; Brown et al., 2022), optimizing their structure for a given task. InstaNAS (Cheng et al., 2020) is an example of input-specific NAS, as it uses a controller that dynamically selects an architecture for each data sample. Meanwhile, AGNAS (Sun et al., 2022) instead utilizes channel-wise attention weights from a shallow MLP to recalibrate concatenated candidate activations before aggregation. Attention weights used to recalibrate channels can be viewed as a channel-wise measures of operation importance.

## 3 METHODS

In this section, we provide a detailed explanation of how we compute the input-specific architecture, as illustrated in Figure 1. We use the same notation as in Section 2.1 for consistency.

### 3.1 INPUT-SPECIFIC ARCHITECTURE

Motivated by the success of attention-based models (Vaswani et al., 2017) and the limitations of standard differentiable NAS (see Section 2.2), we replace static architecture parameters with dynamic parameters computed via patchwise self-attention. This increases expressiveness by adapting the architecture to each input.

**Localized Architecture.** Consider the $k$-th node. Let $F^{(i,j)} \in \mathbb{R}^{B \times C \times H \times W}$ be the activation map obtained by applying the $j$-th operation $o^{(j)}$ to the $i$-th input $x^{(i)}$. We partition each $F^{(i,j)}$ into $P^2$ non-overlapping local patches:

$$F_{u,v}^{(i,j)} \in \mathbb{R}^{B \times C \times \frac{H}{P} \times \frac{W}{P}} \quad \text{for } 1 \leq u, v \leq P. \tag{5}$$

We then apply global average pooling within each patch:

$$\widetilde{F}_{u,v}^{(i,j)} = \frac{P^2}{H \times W} \sum_{p=1}^{\frac{H}{P}} \sum_{q=1}^{\frac{W}{P}} F_{u,v}^{(i,j)}(p,q), \tag{6}$$

yielding $\widetilde{F}_{u,v}^{(i,j)} \in \mathbb{R}^{B \times C \times 1 \times 1}$. This patchwise design lets local features guide architecture choices differently at each spatial location. We refer to this as *mosaic*, since the image-level architecture is composed of patch-level architectures.

In contrast, a single global pooling over the entire activation map can produce overly coarse features, especially in earlier layers that focus on textures and edges. In Section 4.5, we show that the patchwise approach yields more robust and interpretable architecture selection.

**Computing Keys and Query.** We learn *keys* from the pooled features using a shallow two-layer MLP:

$$\boldsymbol{k}_{u,v}^{(i,j)} = K\big(\widetilde{F}_{u,v}^{(i,j)}\big) \in \mathbb{R}^{B \times C}, \tag{7}$$

where

$$\begin{aligned} K(x) &= W_{k_2} \operatorname{LeakyReLU}\big(W_{k_1} x\big), \\ W_{k_1}, W_{k_2} &\in \mathbb{R}^{C \times C}. \end{aligned} \tag{8}$$

Similarly, to obtain a *query*, we concatenate the $(k+1)$ node inputs $\big[x^{(1)}, x^{(1)}, \ldots, x^{(k+1)}\big]$ along the channel dimension:

$$X \in \mathbb{R}^{B \times \big(C \cdot (k+1)\big) \times H \times W}. \tag{9}$$

We again partition $X$ into $P^2$ patches and apply global average pooling to form $\widetilde{X}_{u,v} \in \mathbb{R}^{B \times \big(C(k+1)\big) \times 1 \times 1}$, then map each pooled patch to a query vector:

$$\boldsymbol{q}_{u,v} = Q\big(\widetilde{X}_{u,v}\big) \in \mathbb{R}^{B \times C}, \tag{10}$$

where

$$\begin{aligned} Q(x) &= W_{q_2} \operatorname{LeakyReLU}\big(W_{q_1} x\big), \\ W_{q_1} &\in \mathbb{R}^{C \times \big(C \cdot (k+1)\big)}, \quad W_{q_2} \in \mathbb{R}^{C \times C}. \end{aligned} \tag{11}$$

Table 1: Results on NAS-Bench-201 (Dong & Yang, 2020). We report accuracy of the found architectures on CIFAR-10, CIFAR-100, and ImageNet-16-120.

| METHODS | CIFAR-10 | | CIFAR-100 | | IMAGENET16-120 | |
|---|---|---|---|---|---|---|
| | VALID | TEST | VALID | TEST | VALID | TEST |
| DARTS (LIU ET AL., 2019) | 39.77±0.00 | 54.30±0.00 | 15.03±0.00 | 15.61±0.00 | 16.43±0.00 | 16.32±0.00 |
| GDAS (DONG & YANG, 2019) | 89.89±0.08 | 93.61±0.09 | 71.34±0.04 | 70.70±0.30 | 41.59±1.33 | 41.71±0.98 |
| PC-DARTS (XU ET AL., 2020) | 89.96±0.15 | 93.41±0.30 | 67.12±0.39 | 67.48±0.89 | 40.83±0.08 | 41.31±0.22 |
| IDARTS (ZHANG ET AL., 2021A) | 89.86±0.60 | 93.58±0.32 | 70.57±0.24 | 70.83±0.48 | 40.38±0.59 | 40.89±0.68 |
| DARTS- (CHU ET AL., 2021) | 91.03±0.44 | 93.80±0.40 | 71.36±1.51 | 71.53±1.51 | 44.87±1.46 | 45.12±0.82 |
| AGNAS (SUN ET AL., 2022) | 91.25±0.02 | 94.05±0.06 | 72.4±0.38 | 72.41±0.06 | 45.50±0.00 | 45.98±0.46 |
| $\beta$-DARTS (YE ET AL., 2022) | **91.55±0.00** | **94.36±0.00** | **73.49±0.00** | **73.51±0.00** | **46.37±0.00** | **46.34±0.00** |
| OURS (C10, BEST) | **91.55** | **94.36** | **73.49** | **73.51** | **46.37** | **46.34** |
| OURS (C10) | 91.21±0.68 | 94.21±0.30 | 72.80±1.39 | 72.91±1.20 | 44.97±2.80 | 45.12±2.45 |
| OURS (C100) | 90.20±1.93 | 93.00±1.57 | 70.34±2.13 | 70.58±2.15 | 44.08±2.54 | 43.29±2.45 |
| OPTIMAL | 91.61 | 94.37 | 73.49 | 73.51 | 46.77 | 47.31 |

**Topology-Aware Search Space.** Because DARTS selects two incoming edges per node, we define a topology search space over pairs of candidate edges. For patch $(u, v)$, we consider all valid pairs $((i_1, j_1), (i_2, j_2))$ with $i_1 \neq i_2$. Such reformulation makes decoding straightforward, since finding the top-two candidate edges reduces to finding the top *pair* of candidate edges. We compute an unnormalized attention score:

$$\alpha_{u,v}^{(i_1,j_1,i_2,j_2)} = \frac{\left(\boldsymbol{k}_{u,v}^{(i_1,j_1)} + \boldsymbol{k}_{u,v}^{(i_2,j_2)}\right) \cdot \boldsymbol{q}_{u,v}}{\sqrt{C}}, \tag{12}$$

which indicates the importance of jointly choosing operations $o^{(j_1)}$ and $o^{(j_2)}$ on inputs $x^{(i_1)}$ and $x^{(i_2)}$, respectively, within patch $(u, v)$. In other words, it reflects the importance of selecting node $x_{\text{out}} = o^{(j_1)}(x^{(i_1)}) + o^{(j_2)}(x^{(i_2)})$. We normalize across all valid pairs using softmax:

$$p_{u,v}^{(i_1,j_1,i_2,j_2)} = \frac{\exp\left(\alpha_{u,v}^{(i_1,j_1,i_2,j_2)}\right)}{\sum_{i_1',j_1',i_2',j_2'} \exp\left(\alpha_{u,v}^{(i_1',j_1',i_2',j_2')}\right)}. \tag{13}$$

For stability, we average patch-level distributions to an image-level distribution, producing a single architecture per activation map:

$$p^{(i_1,j_1,i_2,j_2)} = \frac{1}{P^2} \sum_{u=1}^{P} \sum_{v=1}^{P} p_{u,v}^{(i_1,j_1,i_2,j_2)}. \tag{14}$$

By marginalizing over pairs,

$$p^{(i,j)} = \sum_{i',j'} p^{(i,j,i',j')}, \tag{15}$$

the architecture parameters satisfy

$$\sum_{i=1}^{k+1} \sum_{j=1}^{M} p^{(i,j)} = 2, \tag{16}$$

since each pair contributes to two marginals and is therefore counted twice. This condition naturally decodes exactly two edges per node. The node output is then the weighted sum of candidate edges:

$$x_{\text{out}} = \sum_{i,j} p^{(i,j)} F^{(i,j)}. \tag{17}$$

Despite the large combination space, it can be computed efficiently through careful tensor operations rather than explicit iteration, inducing little overhead. See our code for implementation details.

## 3.2 SEARCHING PROTOCOL

Following DARTS, we adopt an approximate bilevel optimization scheme on two disjoint splits of the training data: one split updates the network weights $\omega$ and the other updates the architecture parameters. In MIDAS, the latter include the attention projection networks $\{W_{k_1}, W_{k_2}, W_{q_1}, W_{q_2}\}$ for each node. These parameters are *not* shared across nodes.

Table 2: Results on CIFAR-10 and CIFAR-100 in the DARTS search space. We report architectures discovered on CIFAR-10 and CIFAR-100. For MIDAS, we report mean±std over five independent retraining runs. An asterisk (*) indicates different architectures for CIFAR-10 and CIFAR-100.

| METHODS | GPU (DAYS) | PARAMS (M) | CIFAR-10 | CIFAR-100 |
|---|---|---|---|---|
| NASNet-A (Zoph et al., 2018) | 2000 | 3.3 | 97.35 | 83.18 |
| SNAS (Xie et al., 2019) | 1.5 | 2.8 | $97.15 \pm 0.02$ | 82.45 |
| DARTS (1st) (Liu et al., 2019) | 0.4 | 3.4 | $97.00 \pm 0.14$ | 82.46 |
| DARTS (2nd) (Liu et al., 2019) | 1 | 3.3 | $97.24 \pm 0.09$ | – |
| P-DARTS (Chen et al., 2019) | 0.3 | 3.3 | $97.19 \pm 0.14$ | – |
| PC-DARTS (Xu et al., 2020) | 0.1 | 3.6 | $97.43 \pm 0.07$ | – |
| SDARTS-ADV (Chen & Hsieh, 2020) | 1.3 | 3.3 | $97.39 \pm 0.02$ | – |
| DOTS (Gu et al., 2021) | 0.3 | 3.5 / 4.1* | $\mathbf{97.51 \pm 0.06}$ | $\mathbf{83.52 \pm 0.13}$ |
| DARTS+PT (Wang et al., 2021) | 0.8 | 3.0 | $97.39 \pm 0.08$ | – |
| DARTS- (Chu et al., 2021) | 0.4 | 3.5 / 3.4* | $97.41 \pm 0.08$ | $82.49 \pm 0.25$ |
| AGNAS (Sun et al., 2022) | 0.4 | 3.6 | $97.47 \pm 0.00$ | – |
| $\beta$-DARTS (Ye et al., 2022) | 0.4 | 3.8 | $97.49 \pm 0.07$ | $83.48 \pm 0.03$ |
| Ours (C10) | 0.5 [1] | 3.5 | $97.42 \pm 0.06$ | $83.38 \pm 0.40$ |
| Ours (C100) | 0.5 | 3.3 | $97.25 \pm 0.09$ | $82.67 \pm 0.20$ |

Table 3: ImageNet results when transferring cells discovered on CIFAR-10.

| METHODS | PARAMS (M) | TOP1 (%) | TOP5 (%) |
|---|---|---|---|
| NASNet-A (Zoph et al., 2018) | 5.3 | 74.0 | 91.6 |
| SNAS (Xie et al., 2019) | 4.3 | 72.7 | 90.8 |
| DARTS (Liu et al., 2019) | 4.7 | 73.3 | 91.3 |
| P-DARTS (Chen et al., 2019) | 5.1 | 75.3 | 92.5 |
| SDARTS-ADV (Chen & Hsieh, 2020) | 5.4 | 74.8 | 92.2 |
| DOTS (Gu et al., 2021) | 5.2 | 75.7 | 92.6 |
| DARTS+PT (Wang et al., 2021) | 4.6 | 75.5 | 92.0 |
| $\beta$-DARTS (Ye et al., 2022) | 5.5 | $\mathbf{76.1}$ | $\mathbf{93.0}$ |
| Ours | 5.2 | 75.4 | 92.5 |

**Decoding.** Because MIDAS is input-specific, it does not yield a single fixed set of architecture parameters. After convergence, we therefore need to compute the marginalized parameters $p^{(i,j)}$ over a subset of the training set and average them across samples. The top-two candidate edges $(i, j)$ per node are then selected by their mean importance $\overline{p^{(i,j)}}$. Additional details and an ablation on the decoding subset are provided in Section G.

## 4 EXPERIMENTS

We evaluate MIDAS across multiple search spaces (NAS-Bench-201 (Dong & Yang, 2020), DARTS (Liu et al., 2019), and S1-S4 (Zela et al.)) and three datasets (CIFAR-10, CIFAR-100 (Krizhevsky, 2009), and ImageNet (Deng et al., 2009)) to assess generalizability.

### 4.1 EXPERIMENT DETAILS

**Patch Selection.** We partition each activation map into non-overlapping patches of size $PS \times PS$. For a feature map of height $H$ and width $W$, this yields $P = H/PS = W/PS$ patches (assuming divisibility). If $PS$ is too large, it makes operations harder to discriminate, and an overly small $PS$ can downweight large–receptive-field operations. In the DARTS space we set $PS = 8$ to accommodate larger-context operations such as dilated convolutions. For NAS-Bench-201, which uses only $3 \times 3$ convolutions, we set $PS = 4$. We provide our rationale in Section F.

---

[1] MIDAS incurs ∼1 GPU-hour of additional overhead relative to DARTS (measured on a V100 GPU).

Table 4: Results on S1–S4. Due to the topology-aware search space, S2 and S3 are equivalent for MIDAS.

| DATASET | SPACE | DARTS | RDARTS | DARTS- | DARTS+PT | SHAPLEY-NAS | RF-DARTS | OURS |
|---------|-------|-------|--------|--------|----------|-------------|----------|------|
| CIFAR-10 | S1 | 3.84 | 2.78 | **2.68** | 3.50 | 2.82 | 2.95 | 2.81 |
| | S2 | 4.85 | 3.31 | 2.63 | 2.79 | 2.55 | 4.21 | **2.49** |
| | S3 | 3.34 | 2.51 | **2.42** | 2.49 | **2.42** | 2.83 | 2.49 |
| | S4 | 7.20 | 3.56 | 2.86 | 2.64 | 2.63 | 3.33 | **2.59** |
| CIFAR-100 | S1 | 29.46 | 25.93 | **22.41** | 24.48 | 23.60 | 22.75 | 23.50 |
| | S2 | 26.05 | 22.30 | **21.61** | 23.16 | 22.77 | 22.18 | 22.18 |
| | S3 | 28.90 | 22.36 | **21.13** | 22.03 | 21.92 | 24.67 | 22.18 |
| | S4 | 22.85 | 22.18 | **21.55** | 20.80 | 21.53 | 21.19 | 20.95 |

**Following Best NAS Practices.** To ensure quality and reproducibility, we adhere to recommended NAS practices (Lindauer & Hutter, 2019) and release code for both search and retraining. Aside from a few principled hyperparameter choices, we avoid tuning them not to overfit the benchmark. To prevent cherry-picking, all experiments use four fixed seeds ($\{1, 2, 3, 4\}$). Full results and protocols are reported in Sections B and D.

## 4.2 NAS-BENCH-201 BENCHMARK

NAS-Bench-201 defines a reduced DARTS-like space with four nodes and five operations per edge (15,625 architectures), along with precomputed ground-truth performance on CIFAR-10, CIFAR-100, and ImageNet16-120. Following the standard protocol, we run four searches on CIFAR-10 and report mean±std over four retraining runs using benchmark's provided ground-truth accuracies.

Because NAS-Bench-201 uses a single shared cell, during decoding we average input-specific parameters across cells to obtain one normal and one reduction cell. Results are shown in Table 1. MIDAS attains state-of-the-art performance and finds the globally optimal architecture in three of four runs (see Section B). Although this is uncommon, we also search directly on CIFAR-100 to test robustness. As expected, variance increases with task difficulty, but MIDAS still discovers strong architectures on average.

## 4.3 DARTS SEARCH SPACE

We next evaluate MIDAS in the DARTS search space (Liu et al., 2019). DARTS uses shared normal and reduction cells during both search and retraining. Because MIDAS computes different architectures in each node, to decode MIDAS's input-specific parameters into fixed cells, we adopt the AGNAS (Sun et al., 2022) decoding procedure, which decodes one cell per spatial level, yielding three normal and two reduction cells. See Section G for details and visualizations.

We follow the standard evaluation protocol: conduct four searches, select the supernet with the best validation accuracy, and retrain the derived architecture from scratch for 600 epochs on CIFAR-10 and CIFAR-100. As shown in Table 2, MIDAS is competitive with prior work, reaching $97.42 \pm 0.06\%$ on CIFAR-10 and $83.38 \pm 0.40\%$ on CIFAR-100. Searching directly on CIFAR-100 yields comparable but slightly lower accuracy. To assess transferability, we train the CIFAR-10–discovered cell on ImageNet (Deng et al., 2009). Table 3 again shows competitive results.

## 4.4 ROBUSTNESS OF MIDAS

Zela et al. propose four search spaces S1-S4 to evaluate the robustness of DARTS. We report results on CIFAR-10 and CIFAR-100 in Table 4. Following the RDARTS evaluation protocol, we retrain the best architecture with 20 layers/36 channels on CIFAR-10 and 8 layers/16 channels on CIFAR-100. Since S3 is identical to S2 except for the addition of the *Zero* operation, it is equivalent under MIDAS's topology-aware search space, which does not require *Zero*. Our method achieves state-of-the-art performance across all spaces. In particular, on S2 and S4, MIDAS attains $2.49\%$ and $2.59\%$ test error on CIFAR-10, respectively.

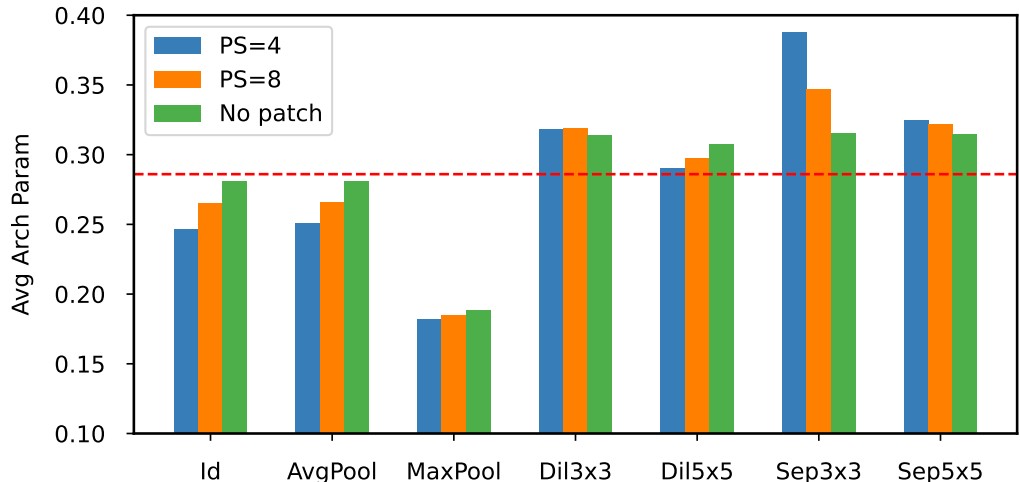

Figure 2: Learned input-specific architecture parameters in the first two cells in the DARTS search space on CIFAR-10, averaged over four runs. We compare three variants: *no patch* (global average pooling only), *PS=4* (patch size $4 \times 4$), and *PS=8* (patch size $8 \times 8$). The horizontal line denotes uniform importance across operations. We observe that *no patch* fails to discriminate among learnable operations, essentially assigning the same weights to all four.

### 4.5 EFFECTIVENESS OF PATCHWISE ATTENTION

We test whether partitioning into patches improves discrimination among candidate operations. Namely, we compare three variants on CIFAR-10: (i) no partition (global average pooling only), (ii) $PS = 4$, and (iii) $PS = 8$. Figure 2 shows the average input-specific parameter for each operation in the first two cells. With $PS = 4$ or $PS = 8$, MIDAS makes consistent, interpretable choices - early cells favour $3 \times 3$ convolutions, which aligns with hand-crafted designs. In contrast, global pooling alone fails to discriminate among learnable operations and assigns near-uniform weights to all four, likely due to overly coarse global features.

### 4.6 ANALYSIS OF INPUT-SPECIFIC ARCHITECTURE

In MIDAS, each architecture parameter is modeled as a distribution over samples rather than a single scalar. As shown in Figure 3, the parameters vary meaningfully across inputs, with variance increasing toward the end of training instead of collapsing into a sharply peaked distribution.

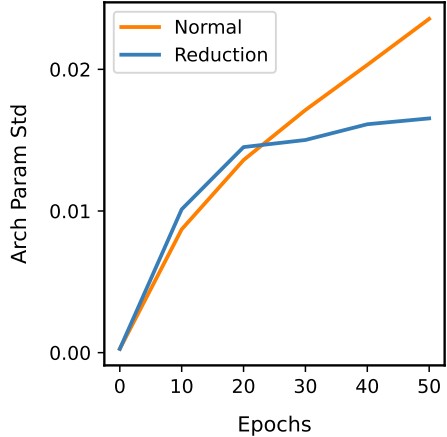

Figure 3: Standard deviation of input-specific architecture parameters $p^{(i,j)}$, averaged over candidate edges, nodes, and four seeds.

**Unimodality.** Ultimately, we must decode one architecture for the entire data distribution. A potential pitfall for MIDAS is *multimodality*, where different samples could produce conflicting architecture decisions, making operation importance hard to quantify. In an extreme case, if a parameter is near $0$ for half the samples and near $1$ for the other half, decoding becomes ambiguous. We therefore test for multimodality using Hartigan's dip test (Hartigan & Hartigan, 1985) on all distributions of architecture parameters (computed over $5000$ samples). In Figure 4 (left), we plot

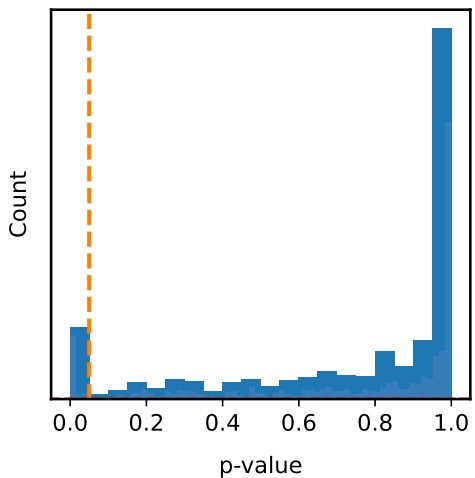 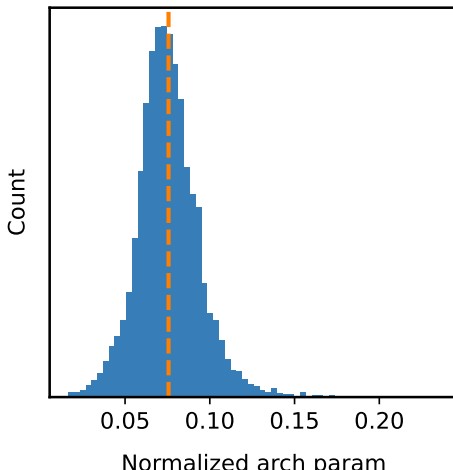

Figure 4: (Left) Histogram of $p$-values from Hartigan's dip test across all architecture parameters. The vertical line marks the $0.05$ rejection threshold. Most $p$-values exceed $0.05$, so unimodality is not rejected for the majority, indicating predominantly unimodal behaviour. (Right) Distribution of an example input-specific parameter, with the vertical line marking the mean value.

the histogram of $p$-values with a $0.05$ rejection threshold. Most $p$-values exceed $0.05$, so we fail to reject unimodality for over $90\%$ of parameters, which indicates predominantly unimodal behaviour. Figure 4 (right) shows an example distribution for a randomly selected operation.

**Class-Aware Architecture.** To test whether these architectural variations reflect noise or structure, in Figure 5 we plot cosine similarities between classes derived from input-specific architecture parameters in the last two cells (details in Section C). We observe that these class similarities align with semantic similarities, indicating that the architecture captures class-aware structure rather than arbitrary variation. This arises from the input-specific nature of MIDAS: if features are similar for related classes, their induced architectures are also similar. Although we do not explore this in our work, this approach could potentially be used to tune architectures for better performance on particularly difficult classes.

## 5 CONCLUSIONS

We introduced MIDAS, a differentiable NAS method that replaces static architecture scalar parameters with input-specific parameters computed via patchwise self-attention, turning architecture selection into an adaptive, local decision. Coupled with a parameter-free, topology-aware search space, MIDAS attains state-of-the-art or competitive results across diverse search spaces.

On NAS-Bench-201, MIDAS consistently discovers the optimal or near-optimal architecture. In the DARTS space, it reaches $97.42\%$ top-1 on CIFAR-10 and $83.38\%$ on CIFAR-100, and transfers to ImageNet with $75.4\%$ top-1. On the RDARTS benchmarks (S1–S4), MIDAS matches or exceeds prior work, achieving $2.49\%$ and $2.59\%$ error on CIFAR-10 in S2 and S4, respectively.

By modernizing DARTS with input-specific self-attention, MIDAS opens a promising new direction for input-specific differentiable NAS. Future work will focus on refining decoding and extending MIDAS to new search spaces and tasks.

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

Table 5: Detailed results of our experiments. **Search Val** is supernet validation accuracy at the end of search. **Retrain** reports params and validation accuracy after retraining the decoded architecture. NAS-Bench-201 validation/test values come from the official tabular benchmark.

| SEARCH SPACE | SEARCH DATASET | SEED | SEARCH VAL (%) | RETRAIN PARAMS (M) | RETRAIN C10 VAL (%) | RETRAIN C100 VAL (%) |
|---|---|---|---|---|---|---|
| DARTS | CIFAR-10 | 1 | 89.51 | – | – | – |
| | | 2 | 89.35 | – | – | – |
| | | 3 | 89.59 | – | – | – |
| | | 4 | 90.07 | 3.5 | 97.42 ± 0.06 | 83.38 ± 0.40 |
| | CIFAR-100 | 1 | 63.57 | – | – | – |
| | | 2 | 64.80 | – | – | – |
| | | 3 | 65.55 | 3.3 | 97.25 ± 0.09 | 82.67 ± 0.20 |
| | | 4 | 64.22 | – | – | – |
| NAS-BENCH-201 | CIFAR-10 | 1 | 80.10 | – | 91.55 | 73.49 |
| | | 2 | 79.24 | – | 91.55 | 73.49 |
| | | 3 | 79.21 | – | 90.20 | 70.71 |
| | | 4 | 80.76 | – | 91.55 | 73.49 |
| | CIFAR-100 | 1 | 45.22 | – | 91.21 | 71.60 |
| | | 2 | 45.67 | – | 87.30 | 67.17 |
| | | 3 | 45.70 | – | 91.08 | 71.00 |
| | | 4 | 44.90 | – | 91.21 | 71.60 |
| S1 | CIFAR-10 | 1 | 89.07 | 3.1 | 97.19 | 76.50 |
| | | 2 | 88.53 | – | – | – |
| | | 3 | 88.91 | – | – | – |
| | | 4 | 88.93 | – | – | – |
| S2 | CIFAR-10 | 1 | 89.81 | 3.9 | 97.51 | 77.81 |
| | | 2 | 89.30 | – | – | – |
| | | 3 | 89.34 | – | – | – |
| | | 4 | 89.51 | – | – | – |
| S4 | CIFAR-10 | 1 | 84.82 | – | – | – |
| | | 2 | 84.36 | – | – | – |
| | | 3 | 85.06 | – | – | – |
| | | 4 | 85.61 | 4.4 | 97.41 | 79.05 |

## A  USE OF GENERATIVE AI

We used an LLM to assist in writing the manuscript and to generate plots. All ideas and experiments were designed and executed by the authors.

## B  EXPERIMENT DETAILS

Table 5 reports per-seed search validation and retraining results across DARTS, NAS-Bench-201, and RDARTS (S1–S4). For NAS-Bench-201, validation/test metrics are taken from the official tabular benchmark and we do not perform any additional retraining.

## C  CLASS-AWARE ARCHITECTURE

To test whether variation in architecture parameters reflects meaningful structure rather than noise, we use class labels as a simple probe. We take a converged CIFAR-10 supernet and compute input-specific architecture parameters for 5000 images (500 per class). For each class, we average these parameters over its 500 samples and restrict the analysis to the last two lowest-resolution cells, yielding a 196-dimensional vector per class (as there are 98 architecture parameters per cell). We z-score each parameter across classes to equalize scales and then compute pairwise cosine similarities, producing the $10 \times 10$ matrix shown in Figure 5.

The matrix shows clear structure. Vehicle classes (airplane, automobile, ship, truck) are mutually similar, while animals form a separate cluster (horse and deer are particularly close). These intuitive patterns suggest that the input-specific parameters capture coherent, class-related structure rather than arbitrary variation.

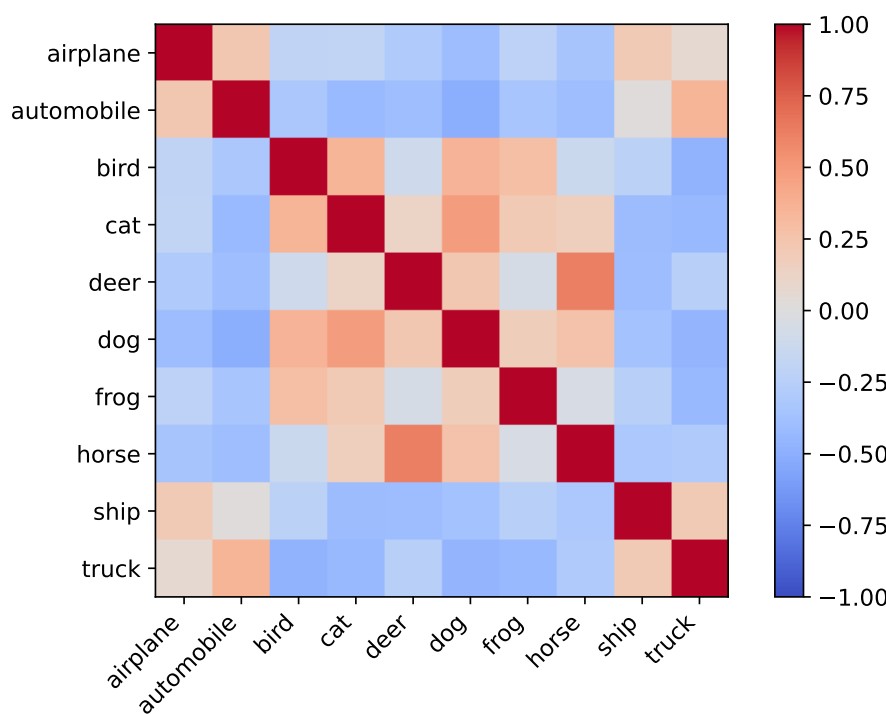

Figure 5: Cosine similarity between CIFAR-10 classes derived from input-specific architecture parameters (higher is more similar). Best viewed in color. See Section C for details.

## D SEARCHING AND RETRAINING PROTOCOLS

We keep DARTS (Liu et al., 2019) default hyperparameters except for three changes during search: (i) weight decay $3 \times 10^{-3}$ (vs. $3 \times 10^{-4}$), (ii) learning rate $10^{-4}$ for attention-based architecture parameters (vs. $10^{-3}$), (iii) cutout size 16. All other settings remain consistent with DARTS and are summarized below.

### D.1 SEARCHING PROTOCOL

During the search phase, DARTS trains a supernet consisting of eight cells, each having four blocks (in the DARTS or RDARTS spaces) or three blocks (in the NAS-Bench-201 space). The second and fifth cells are reduction cells, which downsample the resolution of the activation map. The remaining cells are normal cells. Although standard DARTS employs a single shared normal cell and a single shared reduction cell architecture, our approach computes an input-specific architecture for each cell. The search space includes average pooling (3×3), max pooling (3×3), skip connection, depthwise separable convolution (3×3), depthwise separable convolution (5×5), dilated convolution (3×3), and dilated convolution (5×5). We omit the *Zero* operation, as it is unnecessary in our topology-aware search space.

We search for 50 epochs with a batch size of 64 and $F = 16$ channels, following the bilevel optimization scheme introduced by DARTS (Liu et al., 2019), which alternates gradient updates between convolution weights and architecture parameters. These updates are performed on disjoint halves of the training set to improve generalization. Convolution weights are trained using SGD with momentum 0.9, weight decay $3 \times 10^{-3}$, and a learning rate following a cosine schedule from 0.025 down to $10^{-3}$. Architecture parameters are trained with Adam (standard momentum), weight decay $10^{-3}$, and a learning rate $10^{-4}$. For data augmentation, we employ cutout of size 16. We partition the activation map into patches of size $8 \times 8$ (for DARTS and S1-S4) or $4 \times 4$ (for NAS-Bench-201).

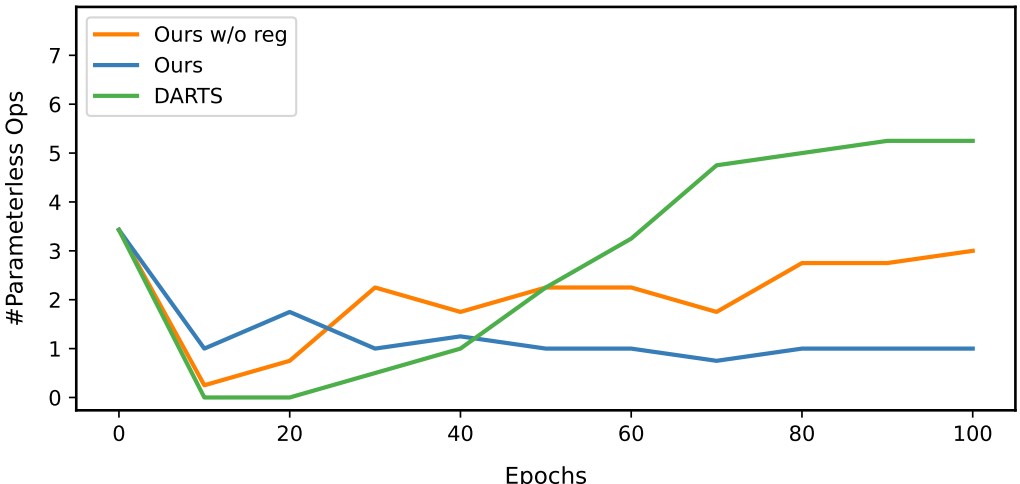

Figure 6: Number of non-learnable operations in the decoded normal cell during search (100 epochs).

## D.2 Retraining Protocol

After decoding the final architecture (detailed in Section G), we retrain decoded architectures with 20 cells, batch size of 96 and $F = 36$ filters. The only exception is the $3N + 2R$ variant in Table 2, which - to ensure fair comparison and that the model has the same number of parameters as other methods - uses $F = 44$. Training runs for 600 epochs using SGD with momentum 0.9, weight decay $3 \times 10^{-4}$, and a learning rate following a cosine schedule from 0.025 down to 0. For data augmentation, we employ cutout of size 16. In line with previous works, we apply drop path, increasing it linearly from 0 to 0.2, and auxiliary towers with weight 0.4. We report the mean of five independent runs per architecture and dataset. For ImageNet retraining, we rely on the SDARTS (Chen & Hsieh, 2020) implementation and refer readers to their paper for additional details.

## E  Stability of Input-Specific Parameters

Replacing scalar $\alpha$ with a shallow attention projection that yields input-specific parameters increases architecture expressiveness and partially balances the bilevel dynamics that can lead to performance collapse. We track the number of non-learnable operations in the decoded normal cell over 100 search epochs (to amplify the instability). Architecture parameters are averaged across normal cells to ensure fair comparison. As shown in Figure 6, DARTS collapses, producing many skip connections, whereas MIDAS remains stable. Weaker regularization introduces more instability than in our default setting, but does not trigger an extreme collapse. Also, this mainly affects pooling which, unlike skip connections, can extract useful high-level features. Nonetheless, we apply stronger regularization to alleviate this behaviour.

## F  Optimal Value of Patch Size

We keep patch size (PS) fixed across spatial resolutions and choose values that divide all CIFAR feature-map sizes (32, 16, 8), which gives $PS \in \{4, 8\}$. This preserves locality while accommodating larger receptive fields. Empirically, we observe that very large PS yields overly coarse features and weak discrimination among operations.

# G  DECODING FINAL ARCHITECTURE

## G.1  DATA SPLIT USED FOR DECODING

We compute input-specific parameters for each candidate edge pair and average over a randomly chosen 10% subset of the training data. To test whether using different subsets or the full training/validation data for decoding is beneficial, we conduct an experiment with 45%, 45% and 10% training set splits. The first two splits are used for weights and architecture training. The 10% split is an empty set to test generalization. Then, we compare architectures (three normal and two reduction cells) decoded using:

- The 45% split that weights were trained on
- The 45% split that architecture was trained on
- The 10% unused split

The three decoded genotypes are identical across all five cells except for a single operation in the final normal cell, where one edge in the third node alternates between a $3 \times 3$ and $5 \times 5$ dilated convolution. In particular, all other four cells are identical among three different decoding approaches. Given such negligible differences, we conclude that it does not impact architecture generalization and we simply reuse DARTS's 50%/50% split without a separate decoding split.

## G.2  DECODING APPROACHES

We select the top-two candidate edges per node using marginalized input-specific parameters averaged over samples. We consider two approaches: (i) average parameters across all normal (respectively reduction) cells to obtain a single genotype for fair comparison, and (ii) AGNAS-style decoding - keep cells #0, #2, #3, #5, #6 and stack as $(6 \times c_0)$, $c_2$, $(6 \times c_3)$, $c_5$, $(6 \times c_6)$. We intentionally avoid excessive tuning to reduce overfitting. In Figures 7 and 8, we visualize the decoded cells for the best CIFAR-10 and CIFAR-100 supernets, respectively.

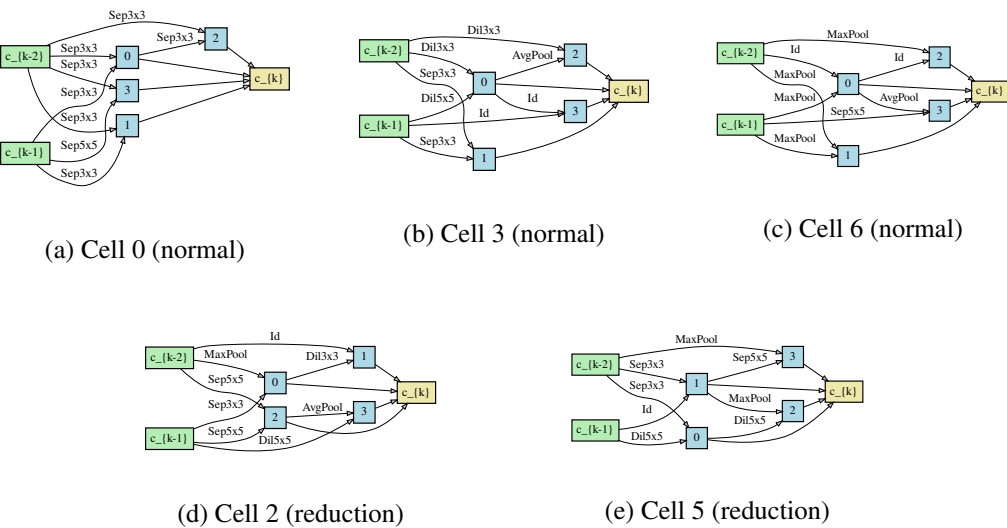

(a) Cell 0 (normal)

(b) Cell 3 (normal)

(c) Cell 6 (normal)

(d) Cell 2 (reduction)

(e) Cell 5 (reduction)

Figure 7: Architecture found by MIDAS on CIFAR-10.

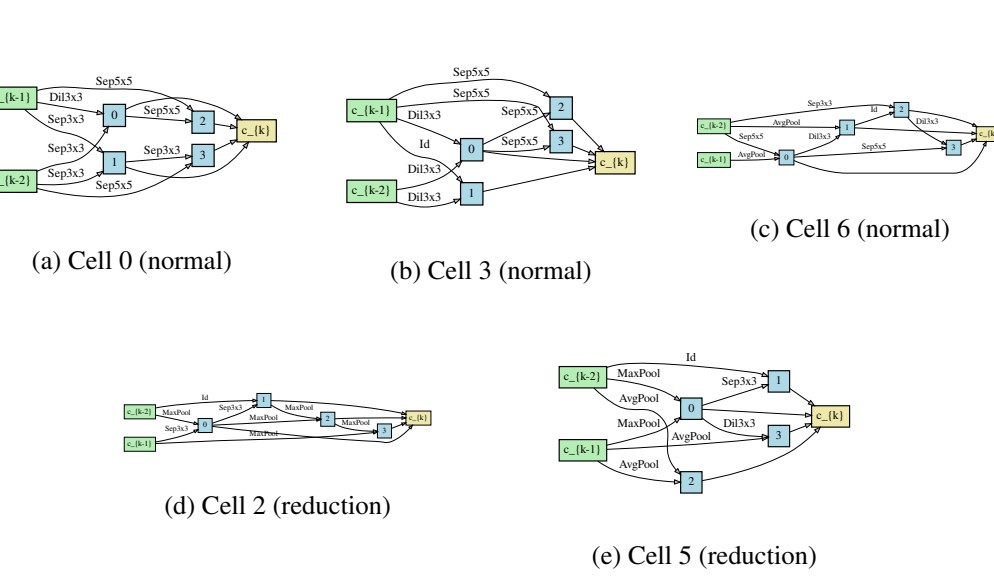

(a) Cell 0 (normal)

(b) Cell 3 (normal)

(c) Cell 6 (normal)

(d) Cell 2 (reduction)

(e) Cell 5 (reduction)

Figure 8: Architecture found by MIDAS on CIFAR-100.

