# OpenReview forum: "MIDAS: Mosaic Input-Specific Differentiable Architecture Search"
_ICLR.cc/2026/Conference — Submitted to ICLR 2026_

### Official Review · Reviewer_19Jq · 2025-10-27

**Soundness:** 2
**Presentation:** 2
**Contribution:** 2
**Rating:** 2
**Confidence:** 4

**Summary:**

The paper proposes MIDAS, a differentiable architecture search (NAS) method that generates input- and position-specific architecture parameters via an attention-based mechanism and a patchwise feature decomposition. It also introduces a “topology-aware” module that scores edge pairs to enforce two-input constraints. The method is evaluated on standard NAS benchmarks like NAS-Bench-201, DARTS, and RDARTS search spaces, reporting competitive results.

**Strengths:**

1.The idea of conditioning architecture parameters on input features is interesting and could make NAS more adaptive.
2.The paper provides extensive experiments and ablations across several standard benchmarks.

**Weaknesses:**

1.I do not see the necessity of partitioning the feature map and learning patch-level architecture parameters, since Eq. (14) simply averages them to obtain a global parameter. This seems mathematically equivalent to directly learning a global weight as in DARTS. The authors should clarify the actual benefit of this design.
2.The use of self-attention is questionable. The defined queries (from raw features) and keys (from operated features) do not form true self-attention, and the semantic meaning of their inner product is unclear. Why should this similarity represent an architecture weight?
3.Compared to DARTS, MIDAS introduces many more architecture parameters and higher computational complexity, but no analysis of FLOPs, memory, or parameter count is provided.
4.The marginal improvement on the DARTS search space is not strong enough to justify the added complexity.

**Questions:**

see Weaknesses.

---

> ### Author Response · Authors · 2025-11-14
>
> Dear Reviewer 19Jq,
>
> Thank you for the review. We address your comments below.
>
> ---
>
> > **1.I do not see the necessity of partitioning the feature map and learning patch-level architecture parameters, since Eq. (14) simply averages them to obtain a global parameter. This seems mathematically equivalent to directly learning a global weight as in DARTS. The authors should clarify the actual benefit of this design.**
>
> The benefit of the patchwise design is both empirical and mathematical, and it is not equivalent to learning a single global weight.
>
> The ablation you ask for is in Section 4.5. In Figure 2, we compare “No patch” (one global parameter) with patch-based variants. The “No patch” setting corresponds exactly to learning one global architecture parameter from a globally pooled feature map. As Figure 2 shows, it assigns almost identical importance to all operations in the first cells, so it fails to distinguish between them and the final architecture is almost random (seed-dependent). In contrast, the patch-based variants produce clear and stable preferences for specific operations.
>
> Averaging patch-level distributions is also mathematically not the same as having one global parameter. In MIDAS, each patch first produces its own normalized distribution over operations and inputs, and then we average these distributions across patches. Due to non-linearity of softmax, the result is different from a global normalization, i.e. `average(softmax(alpha_uv)) != softmax(average(alpha_uv))`.
>
> ---
>
> > **2.The use of self-attention is questionable. The defined queries (from raw features) and keys (from operated features) do not form true self-attention, and the semantic meaning of their inner product is unclear. Why should this similarity represent an architecture weight?**
>
> Our mechanism follows the usual meaning of self-attention: both queries and keys are projections of **the same underlying node inputs**, transformed through different learnable mappings (before vs. after applying a candidate operation). This matches the standard self-attention mechanism where Q and K are learned (linear) projections of **the same input tokens**. The only major change is that the number of keys (#inputs * #operations) and queries (1) is different from the number of input tokens.
>
> The inner product between query and key plays the same role as in transformers: the similarity between a query and key encodes how relevant that key (here: a candidate operation on a specific input) is for the current context. We interpret the resulting normalized scores as **architecture logits** over candidate edges instead of attention logits over tokens, but both are equivalent in terms of representing local architecture weights as similarity.

---

> > ### Author Response · Authors · 2025-11-14
> >
> > > **3.Compared to DARTS, MIDAS introduces many more architecture parameters and higher computational complexity, but no analysis of FLOPs, memory, or parameter count is provided.**
> >
> > We agree that this comparison should be reported explicitly and will add it to the paper.
> >
> > 1. **Search-time overhead (supernet).**
> >    For the DARTS search space on CIFAR-10 (8 cells, 16 channels), we measured:
> >
> >    | Method | Params (M) | FLOPs (M) | Peak GPU mem (MB) | Search time (GPU h) |
> >    | ------ | ---------: | --------: | ----------------: | ------------------: |
> >    | DARTS  |       2.02 |       659 |              6481 |                 4.2 |
> >    | MIDAS  |       2.43 |       673 |              8447 |                 5.2 |
> >
> >    Thus, MIDAS increases:
> >
> >    * parameter count by **~0.41M**,
> >    * FLOPs by **~2%**,
> >    * peak memory by **~2 GB**, and
> >    * search time by **~1 GPU hour** (consistent with the footnote in Sec. 4.1).
> >
> >    The additional cost comes almost entirely from the shallow attention MLPs and patchwise attention, and it is modest compared to the overall DARTS cost.
> > ---
> >
> > > **4.The marginal improvement on the DARTS search space is not strong enough to justify the added complexity.**
> >
> > Our goal is not to make an incremental improvement  to DARTS for a marginal gain on a single benchmark, but to **reformulate differentiable NAS** around input-specific, attention-based architectures with new capabilities and improved robustness. We address your concern on three levels.
> >
> > 1. **Broader empirical picture beyond the DARTS space.**
> >
> >    * **DARTS space (Table 2).** MIDAS reaches **97.42%** on CIFAR-10 and **83.38%** on CIFAR-100, competitive with or close to recent strong methods.
> >    * **NAS-Bench-201 (Table 1).** MIDAS attains **state-of-the-art performance** and finds **the globally optimal architecture** in three of four runs.
> >    * **RDARTS S1–S4 (Table 4).** MIDAS sets the **state of the art** on several robust search spaces, e.g. **2.49%** and **2.59%** test error on CIFAR-10 in S2 and S4, respectively, while remaining strong on CIFAR-100.
> >
> >    Given known reproducibility issues in differentiable NAS (Lindauer & Hutter, 2019; Yang et al., 2020; Yu et al., 2020; Zela et al.), we do our best to follow recommended best practices to provide reproducible and reliable results. We consider a method that is consistently strong and robust across benchmarks more valuable than one that achieves a slightly better single DARTS number at the cost of extensive tuning (and thus benchmark overfitting).
> >
> > 2. **Conceptual novelty**
> >
> >    MIDAS offers several capabilities that DARTS does not:
> >
> >    * **Input-specific architectures.** Each input sample induces its own architecture distribution (p(i,j)), turning architecture parameters from fixed scalars into **distributions over inputs** (Sec. 4.6).
> >    * **Mosaic (patchwise) decisions.** Local patchwise attention allows the architecture to react to local features and yields more discriminative operation selection (Sec. 4.5).
> >    * **Parameter-free topology search.** We jointly model topology (pairs of edges) within the same attention mechanism, avoiding extra parameters as in DOTS while simplifying decoding and improving robustness (Sec. 3.1, Sec. 4.4).
> >    * **New analysis tools.** Because architecture parameters are distributions, we can analyze **unimodality** (Fig. 4) and **class-aware structure** (Fig. 5). For example, we show that architectural similarity between classes aligns with semantic similarity, which is not accessible in standard DARTS.
> >
> > 3. **Complexity vs. benefit.**
> >    As discussed in response to Comment 3, the additional search-time overhead is moderate (~2% FLOPs, ~1 GPU-hour). We believe this is a reasonable cost for:
> >
> >    * a more expressive, input-adaptive formulation,
> >    * improved robustness across multiple search spaces, and
> >    * the ability to analyze and interpret architecture behavior at the distribution level.

---

> > > ### Comment · Reviewer_19Jq · 2025-11-26
> > >
> > > Thank you for your efforts and the detailed clarification. After carefully reviewing your responses, some of my concerns (W1 and W3) have been resolved, so I have decided to raise my score to 4. However, I still believe that the defined queries (from raw features) and keys (from operated features) do not constitute true self-attention, and this design lacks principled justification.

---

### Official Review · Reviewer_ANmx · 2025-10-31

**Soundness:** 3
**Presentation:** 3
**Contribution:** 3
**Rating:** 4
**Confidence:** 4

**Summary:**

This paper introduces the self-attention into the differentiable NAS method. I provides a clear motivation for problems of DARTS that are addressed and the weak correlations stemming from the operation choice in DARTS. input-guided mixing can be a potential fix. Experiments are conducted on two datasets that that show improvements over DARTS baselines.

**Strengths:**

- Interesting idea and novel use of the self-attention technique with a promising new direction. Replacing global architecture parameters with a learned mapping from input features to architecture mixing weights (via attention) is a concrete extension not widely explored in standard DNAS papers.
- Methodological process follows best practices for fairness and reproducibility.

**Weaknesses:**

- The contributions do not read well. There may be made more explicit and clearer.
- Not clear how this scales to larger resolution datesets since the self-attention has quadradic scaling.
- I would like to see more varying datasets used in the experiments for searching for networks. CIFAR100/CIFAR10 have the same properties more or less (e.g., content, resolution)
- Other NAS methods make the problem stricter by incorporating additional constraints with regards to FLOPS/Memory/Parameters
- Limited ablations to understand the impact and main mechanisms providing that improve performance

**Questions:**

- Can you provide more connections to the dynamic routing literature?
- Why are the results between proposed approach and β-DARTS in Table 1 are exactly the same?

---

> ### Author Response · Authors · 2025-11-14
>
> Dear Reviewer ANmx,
>
> Thank you for the review. We address your comments below.
>
> > **The contributions do not read well. There may be made more explicit and clearer.**
>
> We will revise the introduction to clearly list our contributions as:
>
> 1. **Input-specific differentiable NAS:** We introduce MIDAS, which replaces static DARTS architecture scalars with input-specific parameters computed via patchwise self-attention over (input, operation) pairs, turning architecture selection into a dynamic, per-input mechanism.
> 2. **Mosaic (patchwise) architecture:** We propose a patchwise “mosaic” formulation that makes local architecture decisions, improving discrimination between candidate operations and alleviating rank disorder.
> 3. **Parameter-free topology search:** We integrate topology (edge-pair) search into the same attention mechanism without introducing extra parameters, simplifying decoding and avoiding additional hyperparameters.
> 4. **Robust empirical behaviour and analysis:** MIDAS achieves state-of-the-art or competitive results on NAS-Bench-201, DARTS, and RDARTS (S1-S4), and we provide analysis (patchwise vs global, unimodality, class structure) that helps explain *why* it works.
>
> > **Not clear how this scales to larger resolution datasets since the self-attention has quadradic scaling.**
>
> MIDAS does **not** apply self-attention over all spatial locations. Instead, it treats **(input, operation)** pairs as tokens. The complexity of the attention is therefore:
>
> * quadratic in the *number of tokens per patch* = (number of inputs to the node) x (number of operations), which is small and fixed by the DARTS-style search space, and
> * **linear** in the number of patches (hence linear in image resolution for a fixed patch size).
>
> In other words, self-attention in MIDAS is *not* quadratic in HxW - increasing resolution only increases the number of independent patches, not the token set on which we perform pairwise attention.
>
> We will clarify this complexity explicitly in the paper.
>
> > **I would like to see more varying datasets used in the experiments for searching for networks. CIFAR100/CIFAR10 have the same properties more or less (e.g., content, resolution)**
>
> We follow the standard evaluation protocol used in differentiable NAS:
>
> * For **search**, we use CIFAR-10 or CIFAR-100 in the DARTS and RDARTS spaces, and CIFAR-10/CIFAR-100 for NAS-Bench-201.
> * For **evaluation**, we report results on CIFAR-10, CIFAR-100, ImageNet16-120 (via NAS-Bench-201), and ImageNet (via transfer from CIFAR-10 cells).
>
> In addition, we use multiple **search spaces** (DARTS, NAS-Bench-201, S1-S4 RDARTS), which differ significantly in topology and operation sets. This follows common practice in the field and provides a reasonably comprehensive test suite for differentiable NAS methods.
>
> For reference, many compared methods use essentially a subset of our suite of search spaces:
>
> | Method    | DARTS | NAS-Bench-201 | RDARTS | MobileNet-like |
> | --------- | :---: | :-----------: | :----: | :------------: |
> | $\beta$-DARTS   |   ✓   |       ✓       |    –   |        –       |
> | AGNAS     |   ✓   |       ✓       |    –   |        ✓       |
> | DARTS-    |   ✓   |       ✓       |    ✓   |        –       |
> | iDARTS    |   ✓   |       ✓       |    –   |        –       |
> | DARTS-PT  |   ✓   |       ✓       |    –   |        –       |
> | FairDARTS |   ✓   |       –       |    –   |        ✓       |
> | RDARTS    |   ✓   |       –       |    ✓   |        –       |
>
> MobileNet-like (ImageNet) search spaces are less standardized. However, if feasible within the discussion period, we will conduct experiments there and report the results.

---

> > ### Author Response · Authors · 2025-11-14
> >
> > > **Other NAS methods make the problem stricter by incorporating additional constraints with regards to FLOPS/Memory/Parameters**
> >
> > This line of work targets **hardware-aware or multi-objective NAS**, which we view as a related but distinct subfield. Our focus in MIDAS is on **improving the modeling of architecture parameters and robustness within the standard differentiable NAS framework**, not on optimizing FLOPs/latency/parameters under explicit constraints.
> >
> > We do not contribute nor claim to contribute to hardware-aware NAS. However, MIDAS is compatible with such constraints and could be combined with hardware-aware objectives in future work.
> >
> > > **Limited ablations to understand the impact and main mechanisms providing that improve performance**
> >
> > We do include targeted ablations and analysis of the main mechanisms we introduce:
> >
> > * **Patchwise vs global aggregation.** Section 4.5 compares global pooling (“no patch”) with patch sizes PS=4 and PS=8, showing that patchwise attention yields much stronger discrimination among operations, while global pooling collapses towards near-uniform weights.
> > * **Unimodality of architecture distributions** We analyze the learned architecture distributions and show that MIDAS tends to produce unimodal, well-separated architecture decisions.
> > * **Class-aware structure.** We analyze class-conditioned architecture parameters and observe meaningful class structure (e.g. similar classes sharing similar architecture patterns), which supports our claim that input-specific parameters capture non-trivial information.
> > * **Stability and regularization.** Appendix E analyzes stability of the learned architectures versus DARTS, including a weaker-regularization variant of MIDAS, showing that input-specific attention mitigates collapse into skip connections.
> > * **Decoding robustness.** Appendix G compares different decoding subsets and shows that the decoded genotypes are nearly identical, indicating that the decoding procedure is stable.
> >
> > We agree that more exhaustive ablations (e.g. alternative attention architectures or query/key parameterizations) are always possible. Within page limits, we chose experiments that directly target the mechanisms that distinguish MIDAS from prior DARTS-style methods.
> >
> > > **Can you provide more connections to the dynamic routing literature?**
> >
> > Thank you for this comment. Dynamic routing is typically studied in the context of **input-dependent selection of computation paths** (e.g. mixture-of-experts or instance-aware NAS), which is conceptually related to our input-specific architecture selection.
> >
> > However, MIDAS operates in a different regime: we **stay inside the DARTS-style weight-sharing supernet and search space**, and we use self-attention over (input, operation) pairs to produce architecture parameters, rather than routing between separate expert networks. We will add a short paragraph in the related work section to clarify this connection and contrast MIDAS with dynamic routing and input-specific NAS (e.g. InstaNAS).
> >
> > > **Why are the results between proposed approach and $\beta$-DARTS in Table 1 are exactly the same?**
> >
> > This is a good observation. NAS-Bench-201 is a **tabular** benchmark with a finite set of 15,625 architectures and precomputed accuracies. Both $\beta$-DARTS and MIDAS independently discover (most likely) **exactly the same architecture**, which happens to be the globally optimal one in the table. Since evaluation is done by *looking up* the precomputed accuracy rather than retraining, identical discovered architectures naturally yield identical numbers.

---

> > > ### Comment · Reviewer_ANmx · 2025-11-27
> > >
> > > Thank you very much for your feedback and clarifications. I am happy that a lot of the points have been addressed and encourage the authors to make the necessary clarifications and revisions in the main manuscript as well. I am happy to raise my score to 6.

---

### Official Review · Reviewer_Jzrv · 2025-11-05

**Soundness:** 2
**Presentation:** 2
**Contribution:** 2
**Rating:** 2
**Confidence:** 4

**Summary:**

This paper presents a variant of DARTS. Using self-attention mechanism to design a architecture selection process. Experiments have been done on common toy search space like DARTS, NASBench201 and RDARTS on image classification task.

**Strengths:**

Paper is written pretty clear about their technical detail.
They put three space and provide some ablation on their own method.

**Weaknesses:**

Weaknesses:

As using differentiable archtiecture search is try to minimize the search cost, this field still lacks certain thereotical support, why using a shared weights super net and differentiable target towards certain architecture selection process can work. Any research without clearly addressing direction seems to be meaningful only on practical point of view.

In this regard, this paper presents a result that only on-par or even worse compared to some old baselines, like beta-DARTS in 2022.

The only motivation on this one is to leverage the full representation power of self-attention mechanism, which is quite straightforward and trivial honestly speaking.

As such, I have not seen any reason to accept this paper, for lacking interesting motivation, compelling result.

**Questions:**

As weaknesses

---

> ### Author Response · Authors · 2025-11-14
>
> Dear Reviewer Jzrv,
>
> Thank you for the review. We respond point by point below.
>
> > **Experiments have been done on common toy search space like DARTS, NASBench201 and RDARTS on image classification task.**
>
> We use DARTS, NAS-Bench-201, and RDARTS because they form the standard evaluation suite for differentiable NAS. NAS-Bench-201 is an exhaustive, tabular benchmark. RDARTS was designed to test robustness of DARTS-like methods. DARTS remains the canonical search space in this line of work. We would therefore not describe them as “toy”, but rather as the main community benchmarks.
>
> > **As using differentiable architecture search is try to minimize the search cost, this field still lacks certain thereotical support, why using a shared weights super net and differentiable target towards certain architecture selection process can work. Any research without clearly addressing direction seems to be meaningful only on practical point of view.**
>
> We agree that differentiable NAS (and weight-sharing NAS more broadly) is still primarily an empirical field. Section 2.1 provides preliminaries where we restate the DARTS framework and its rationale, and we explicitly build on this line of work rather than proposing an entirely new optimization principle. For further discussion of why a shared-weight supernet and a continuous relaxation can work, we refer to the original DARTS paper and follow-up analyses, which provide precisely the kind of theoretical support you are asking for.
>
> We are somewhat surprised to see this presented as a specific weakness of our paper. The lack of a complete theory is a property of differentiable NAS as a whole, not something introduced by MIDAS. If one were to treat this as a decisive flaw, this would invalidate most of the DARTS literature and, arguably, a large fraction of deep learning research, which is often driven by empirical work with only partial theoretical understanding. Our contribution should be judged within this established differentiable NAS framework: we keep the same basic principles as DARTS and improve the modeling of architecture parameters via self-attention and input-specific distributions.
>
> > **In this regard, this paper presents a result that only on-par or even worse compared to some old baselines, like beta-DARTS in 2022.**
>
> We respectfully disagree with this comment. We compare MIDAS against a broad set of methods, including both older and newer baselines, across several search spaces. The fact that many strong baselines date from 2019-2022 reflects the history of the field: most progress on DARTS-like methods happened then, and it has slowed significantly since due to severe fragility and reproducibility issues.
>
> As detailed in the main paper, MIDAS is competitive on the DARTS space, achieves state-of-the-art results on several RDARTS settings, and in NAS-Bench-201 it consistently finds the globally optimal architecture. We prefer to report results that are reproducible and robust across multiple benchmarks rather than inflate them to surpass every single number in the literature. We strongly believe that such a lack of scientific rigor was one of the reasons behind the stagnation of the field.
>
> > **The only motivation on this one is to leverage the full representation power of self-attention mechanism, which is quite straightforward and trivial honestly speaking.**
>
> We respectfully disagree with this comment. Our goal is not simply to “plug in self-attention”, but to rethink how architecture parameters are represented and used. MIDAS turns fixed architecture scalars into input-specific, patchwise distributions (and also unifies operation and topology selection in a single attention mechanism, or enables new architecture analysis). While the individual building blocks (self-attention, patch pooling) are of course familiar, their combination to redesign differentiable NAS is neither trivial nor purely cosmetic, to say the least.
>
> Differentiable NAS has accumulated many incremental variations while adopting few such conceptual shifts. If such a change is deemed “straightforward and trivial honestly speaking”, then we can only express our admiration that the reviewer has the intellectual capability and creativity to regard such a reformulation as straightforward.
>
> > **As such, I have not seen any reason to accept this paper, for lacking interesting motivation, compelling result.**
>
> We respectfully disagree with this overall assessment. MIDAS (i) builds directly on the standard differentiable NAS framework, (ii) introduces a novel attention-based, input-specific parameterization of architecture parameters, and (iii) achieves competitive or state-of-the-art performance on DARTS, RDARTS, and NAS-Bench-201 under robust, reproducible protocols. Taken together, we believe this provides both a clear motivation and a compelling empirical contribution to the differentiable NAS literature.

---

### Meta-Review · Area_Chair_cTHv · 2026-01-07

**Summary:**

I agree with Reviewer Jzrv that current NAS research lacks sufficient theoretical support. I also share Reviewer 19Jq’s concerns regarding the use of self-attention. In particular, the semantic meaning of the inner product in this setting is unclear, and it is not obvious why it should be interpreted as an architecture weight. A clear justification of this design choice—preferably supported by theoretical analysis—is necessary to establish the soundness of the proposed method.


The concern is the fundamental effectiveness of the proposed approach. There is a growing view that the community’s heavy emphasis on Neural Architecture Search (NAS) may, in some cases, hinder progress by diverting effort away from human-driven, handcrafted design. Historically, the most significant architectural advances—including those underlying modern large language models—have emerged from expert-designed architectures rather than automated search. The effectiveness of NAS has long been questioned by some people. Against this backdrop, any new NAS method must demonstrate clear and compelling advantages to justify its practical value.

A related and more specific concern is the reliability of NAS methods. Although the authors briefly mention the issue of ranking disorder, their discussion remains superficial. The core problem is not merely the misranking of individual operations, but whether the method can reliably evaluate entire architectures. To illustrate this concern in the context of DARTS-style approaches and the proposed method: if we select multiple complete architectures, aggregate the learned weights of their constituent operations, and rank architectures according to these aggregated scores, does this ranking correlate with their true test performance? I am not convinced that such a correlation holds in general. The authors should provide empirical evidence demonstrating that the aggregated weights of a full architecture are predictive of its actual performance. Without validation at the architectural level—rather than the operation level—the claimed effectiveness remains unsubstantiated.

Alternatively, the authors could strengthen their claims through rigorous theoretical analysis. In the absence of either empirical validation on complete architectures or a solid theoretical foundation, the robustness of the results is questionable.

**Reviewer Concerns:**

While the authors have partially addressed the empirical inquiries, particularly those raised by Reviewer ANmx, the most critical concerns remain unresolved. specifically, I align with Reviewer Jzrv regarding the lack of sufficient theoretical support in this line of research. Furthermore, Reviewer 19Jq’s critique regarding the semantic meaning of the inner product is significant; it is not intuitively obvious why this similarity metric should be interpreted as an architecture weight. A rigorous justification for this design choice—ideally supported by theoretical analysis—is strictly necessary to establish the soundness of the proposed method.

**Reviewer Scores:**

Assuming there are no issues related to unfair evaluation, Reviewers Jzrv and 19Jq may not substantially change their scores (e.g., from reject to accept). However, the situation changes if those unfair issues are taken into account. Meanwhile, Reviewer ANmx appears open to reconsidering.

---

### Decision · Program_Chairs · 2026-01-26

Reject